# Genomic characterisation of Eµ-*Myc* mouse lymphomas identifies *Bcor* as a *Myc* co-operative tumour-suppressor gene

Marcus Lefebure[1,2,*], Richard W. Tothill[1,2,3,*], Elizabeth Kruse[1], Edwin D. Hawkins[4], Jake Shortt[1,5], Geoffrey M. Matthews[6], Gareth P. Gregory[1], Benjamin P. Martin[1], Madison J. Kelly[1], Izabela Todorovski[1], Maria A. Doyle[1], Richard Lupat[1], Jason Li[1], Jan Schroeder[4], Meaghan Wall[7,8], Stuart Craig[1], Gretchen Poortinga[1], Don Cameron[1], Megan Bywater[1], Lev Kats[1,2], Micah D. Gearhart[9], Vivian J. Bardwell[9], Ross A. Dickins[10], Ross D. Hannan[1,11], Anthony T. Papenfuss[1,2,4] & Ricky W. Johnstone[1,2]

The Eµ-*Myc* mouse is an extensively used model of MYC driven malignancy; however to date there has only been partial characterization of MYC co-operative mutations leading to spontaneous lymphomagenesis. Here we sequence spontaneously arising Eµ-*Myc* lymphomas to define transgene architecture, somatic mutations, and structural alterations. We identify frequent disruptive mutations in the PRC1-like component and BCL6-corepressor gene *Bcor*. Moreover, we find unexpected concomitant multigenic lesions involving *Cdkn2a* loss and other cancer genes including *Nra*s, *Kras* and *Bcor*. These findings challenge the assumed two-hit model of Eµ-*Myc* lymphoma and demonstrate a functional *in vivo* role for *Bcor* in suppressing tumorigenesis.

[1] The Peter MacCallum Cancer Centre, Melbourne, Victoria 3000, Australia. [2] The Sir Peter MacCallum Department of Oncology, University of Melbourne, Parkville, Victoria 3052, Australia. [3] Department of Pathology, University of Melbourne, Parkville, Victoria 3052, Australia. [4] The Walter Eliza Hall Institute of Medical Research, Parkville, Victoria 3052, Australia. [5] School of Clinical Sciences at Monash Health, Faculty of Medicine, Nursing & Health Sciences, Clayton, Victoria 3168, Australia. [6] Dana Farber Cancer Institute, Boston, Massachusetts 02115, USA. [7] Victorian Cancer Cytogenetics Service, St Vincent's Hospital, Fitzroy, Victoria 3065, Australia. [8] Department of Medicine, St Vincent's Hospital, University of Melbourne, Parkville, Victoria 3052, Australia. [9] Developmental Biology Center, Masonic Cancer Center, and Department of Genetics, Cell Biology, and Development, University of Minnesota, Minneapolis, Minnesota 55455, USA. [10] Australian Centre for Blood Diseases, Monash University, AMREP Building, Commercial Road, The Alfred Hospital, Melbourne, Victoria 3004, Australia. [11] Cancer Biology and Therapeutics Department, John Curtin School of Medical Research, Australian National University, Canberra, Australian Capital Territory 0200, Australia. * These authors contributed equally to this work. Correspondence and requests for materials should be addressed to R.W.T. (email: Richard.Tothill@petermac.org) or to R.W.J. (email: Ricky.Johnstone@petermac.org).

Genetically engineered mouse models (GEMMs) have been extensively used to validate oncogene or tumour suppressor function and serve as tractable models for pre-clinical evaluation of new therapeutics[1]. Reminiscent of human malignancies, considerable disease heterogeneity can be observed in GEMMs despite the mostly inbred homozygous genotype and their controlled housing environment[2]. Similar to humans, inter-tumoural heterogeneity in GEMMs is likely caused by additional somatic lesions that occur spontaneously and are required for tumorigenesis. GEMMs in conjunction with high-throughput genomics can therefore be used as 'discovery' resources to identify and validate spontaneous co-operating mutations and thus providing important information on the genetic interactions that underpin tumour onset and progression[3–6].

The Eμ-*Myc* mouse model employs a transgene mimicking the t(8:14) translocation of c*Myc* and *IgH* enhancer elements characteristic of human Burkitt lymphoma[7]. Eμ-*Myc* mice develop B-cell leukaemia/lymphoma like malignancy with 100% penetrance. Disease in these mice typically manifests as a period of premalignant polyclonal precursor-B cell expansion followed by self-limiting remission and onset of clonal lymphoid neoplasia ranging from a pre-B to naive B phenotype[8,9]. Numerous articles have been published using this model to interrogate the efficacy and mechanisms of action of diverse anti-cancer agents and to study putative functional interactions between *Myc* and candidate cancer causing genes. For example, a landmark study by Strasser and colleagues using the Eμ-*Myc* model provided definitive evidence that MYC and BCL2 can functionally cooperate to accelerate lymphoma development[10]. In addition, the role of the p19ARF-MDM2-p53 axis in regulating MYC-mediated apoptosis and lymphomagenesis has been comprehensively dissected at the genetic, biochemical and biological level using the Eμ-*Myc* model[11,12]. Oncogenic RAS can also have an important co-operative role in MYC transformation and spontaneous activating *Nras* mutations were identified in Eμ-*Myc* lymphomas[13]. Taken together, these studies highlight the utility of the model as a cancer gene-discovery resource and its extensive application for accurately studying the biology of the *Myc* oncogene.

Despite Eμ-*Myc* lymphomas being initiated by a single oncogene on a defined genetic background, there is a remarkable degree of inter-tumour heterogeneity, reflected by gene expression profiling and the latency of disease onset[14]. This is consistent with studies indicating that Eμ-*Myc* lymphomas arise through acquisition of secondary or tertiary mutations that de-regulate tumour suppressor pathways such as those mediated by p19ARF and p53 that normally counteract the oncogenic effects of MYC. The intrinsic apoptotic pathway is clearly implicated in Eμ-*Myc* lymphomagenesis. Overexpression of *Bcl2*, forced expression of other pro-survival BCL2 family members (for example, *Bclxl*, *Mcl1*) or deletion of pro-apoptotic BH3-only genes (for example, *Bim*, *Bmf*, *Bad*) can accelerate lymphoma development in the Eμ-*Myc* mouse[15–18]. However, whether mutational dysregulation of the BCL2 family actually occurs in spontaneously arising tumours is not yet defined. Disruption of non-apoptotic tumour suppressive mechanisms are also implicated in Eμ-*Myc* lymphomagenesis, in particular oncogene induced senescence pathways[19,20] and immune surveillance[21]. Despite clear evidence that naturally occurring mutations in *Trp53*, *Cdkn2a* and *Nras* co-operate with MYC in the Eμ-*Myc* model, other co-operating *de novo* genetic lesions remain unknown in up to half of all cases[12]. In particular, it is important to determine if disruption of genes controlling disparate apoptotic and senescence pathways that can co-operate with MYC, when synthetically de-regulated are actually affected in spontaneously arising Eμ-*Myc* lymphomas. We therefore used massively-parallel sequencing to catalogue genetic lesions arising in Eμ-*Myc* lymphomas.

We found that in addition to the expected mutations or deletions in *Trp53*, *Cdkn2a* and *Nras*, deleterious mutations in BCL6-corepressor gene *Bcor* frequently occurred in Eμ-*Myc* lymphomas. Subsequent functional studies confirmed the tumour suppressor role of *Bcor*, and Eμ-*Myc* lymphomas with experimental depletion or deletion, or spontaneous mutation of *Bcor* presented with a unique gene expression signature indicating that TGFβ signalling was aberrant in these lymphomas. Finally, we discovered that Eμ-*Myc* lymphomas co-occur with loss of *Cdkn2a* and either activating mutations in *Ras* or deleterious mutations in *Bcor* providing evidence that loss of *Cdkn2a* alone may not be sufficient to cooperate with overexpressed *Myc* to drive lymphomagenesis.

## Results

**Transgene architecture in Eμ-*Myc* lymphoma.** To comprehensively characterize the genetic architecture of the Eμ-*Myc* transgenic mouse, we applied whole-genome sequencing (WGS) to a spontaneous Eμ-*Myc* lymphoma (#88) and its matching germline DNA extracted from tail tissue of a hemizygous transgenic animal. Augmenting the mouse reference genome with the pUC12 vector sequence of the Eμ-*Myc* transgene enabled mapping of break points within the transgene and enumeration of transgene copy number in germline and lymphoma (Fig. 1a). We identified all the expected elements of the transgene cassette including the cloned translocation between chr12 (*IgH* enhancer) and chr15 (*Myc*) in addition to the flanking cloning sites within the pUC12 vector DNA (ref. 7). A single breakpoint was detected between chr19 and pUC12 vector sequence 5′ of the Eμ-*Myc* transgene. FISH analysis using fluorescently labelled BAC probes confirmed juxtaposition of *Myc* and *Jak2* (located on chr19) proximal to transgene insertion on chr19, which is consistent with observations from a previous study (Supplementary Fig. 1)[22].

Copy-number analysis of WGS data revealed that five copies of *Myc* were present in the hemizygous Eμ-*Myc* germline, corresponding to three transgene copies and two endogenous copies of *Myc*. We also observed a copy-number gain of a ∼3 Mb segment of chr19 proximal to site of transgene insertion (Fig. 1b), as previously reported[22]. The chr19 segment plus one extra copy of the Eμ-*Myc* transgene underwent further somatic gain (+1) in the #88 Eμ-*Myc* lymphoma, which has not been previously reported. Additional mate-pair WGS indicated that the transgene copies are likely arranged as concatenated repeats (Fig. 1b).

We confirmed germline amplification of the chr19 segment in Eμ-*Myc* transgenic mice bred at two additional institutions (Walter and Eliza Hall Institute (WEHI) Melbourne Australia, Cold Spring Harbor Laboratories USA) (Supplementary Fig. 2). This indicates that these structural arrangements are almost certainly founding events that occurred following the pronuclear injection of the transgenic vector. The amplicon on murine chr19 is syntenic to the human 9p24.1 region, which is frequently amplified in Hodgkin's Lymphoma (HL) and primary mediastinal B-cell lymphoma (PMBCL) and contains the tumour-promoting genes *JAK2* and *CD274* (PD-L1)[23]. Fusello *et al.*[22] previously showed that pre-malignant and transformed B-cells from Eμ-*Myc* mice exhibited higher PD-L1 expression compared with wild type B-cells, concluding that this was causally related to the chr19 amplification. However, we found that PD-L1 protein expression did not correlate with gene dosage inferred from additional somatic gain of the chr19 amplicon (Supplementary Fig. 3). We also found that Eμ-*Myc* lymphomas did not show appreciable activation of JAK2 signalling as assessed by phosphorylation of the downstream target STAT5 and furthermore, JAK2 inhibition using FDA-approved small molecule inhibitor ruxolitinib had no effect on survival of

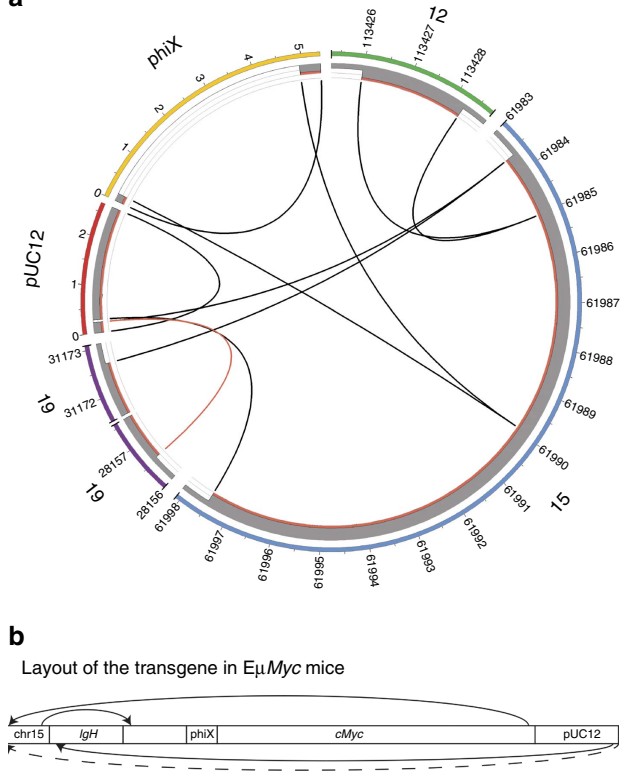

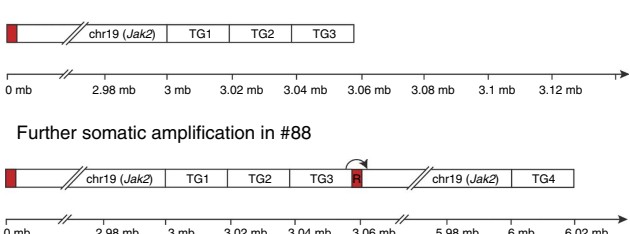

**Figure 1 | Transgene architecture in Eµ-*Myc* lymphoma. (a)** Circos plot showing the Eµ-*Myc* transgenic cassette (pUC12, phiX, chr12 (Eµ), chr15 (*Myc*), identified breakpoints and estimated copy number. Outer coloured bars depict chromosome and transgene segments. Grey bars and links represent germline copies with two copies per increment. Red bars represent additional somatic copy-number gain and breakpoints. **(b)** Schematic showing the arrangement of the Eµ-*Myc* transgene (upper), the three repeats (TG1-3) of the transgene in the Eµ-*Myc* germline (middle) and the extra gain of the transgene and the segment of chr19 in lymphoma #88 (lower). This figure was adapted from Adams *et al.* (ref. 7). R in red band indicates repetitive DNA elements.

Eµ-*Myc* lymphomas cultured *in vitro* (Supplementary Fig. 4). This suggests germline or somatic amplification of *Jak2* is unlikely to contribute to initiation and maintenance of the Eµ-*Myc* lymphomas. The role of other genes within the amplified chr19 region (for example, *Il33*, *Pdcd1lg2*) (Supplementary Data 1) remains to be clarified and is the subject of ongoing experiments.

**WES identifies new gene drivers of Eµ-*Myc* lymphoma.** We next applied whole-exome sequencing (WES) to explore the number, type and frequency of somatic mutations in 23 spontaneous Eµ-*Myc* lymphomas (Supplementary Data 2). Sixteen cases were derived prospectively and seven cases were taken from retrospectively archived lymphomas. WES of normal tail DNA from Eµ-*Myc* animal #88 and a littermate control from

the prospective series (ML62) was used to filter germline polymorphisms from the data. Furthermore, dbSNP annotated variants or any single nucleotide variants (SNV) or insertion/deletion (InDel) recurring two or more times in closely related animals were removed from analysis as these variants were most likely polymorphisms or recurrent sequencing artifacts. A subset of SNVs and InDels were further validated using targeted amplicon-based sequencing confirming the high specificity of 92.6% (75/81) of our variant calling pipeline. Finally, to confirm the somatic driver mutations (described below) we sequenced matching isolated circulating B-cells from individual mice at 4-weeks of age showing undetectable or very low-variant allele frequency (VAF) compared with VAF measured in blood at disease end point (Supplementary Fig. 5).

The lymphomas from the prospective cohort each had 3–31 SNVs or InDels predicted to truncate or alter the translated amino acid sequence of proteins encoded by annotated genes (Fig. 2a). As expected we detected *Trp53* ($n = 4$) and *Nras* ($n = 2$) mutations. In addition we identified *Kras* ($n = 4$) mutations that have not been previously described in Eµ-*Myc* lymphomas (Fig. 2b). The most frequently mutated gene was the BCL6-copressor (*Bcor*), recurrently mutated in seven lymphomas (32%) with either frameshift InDels or nonsense mutations predicted to cause premature protein truncation. A splice-site variant in the PRC2 complex subunit *Ezh2*, a gene previously shown to function as a tumour-suppressor in the Eµ-*Myc* model[24], was observed in one case (ML33). Lymphoma ML39 harboured a mutation in the gene *Mtor* affecting a residue conserved between mice and humans and predicted to be damaging using SIFT algorithm. Deregulation of the AKT-MTOR-eIF4E pathway has been shown to co-operate with MYC in Eµ-*Myc* lymphomagenesis[25] and treatment of Eµ-*Myc* transgenic mice with mTORC1 inhibition by everolimus during the pre-malignant stage of disease significantly delayed lymphoma onset[20]. Activating *MTOR* mutations have also previously been described in several human cancer types including diffuse large B-cell lymphoma[26,27]. Lymphoma #88 harboured a mutation in *mir142* that has been reported in ~20% of diffuse large B-cell lymphomas[28]. Lymphomas ML352 and #299 had mutations in the ribosomal protein *Rpl10*, a gene found recurrently mutated in T-ALL (ref. 29). Three lymphomas (#218, #219, ML20) did not harbour SNVs or InDels in genes that have any obvious link to cancer.

We analysed WES ($n = 9$) and low to medium coverage WGS data ($n = 13$) (unpaired) to detect somatic copy-number alterations (SCNAs) in lymphoma samples. Focal deletion events of *Cdkn2a* were detected in five cases, validated by quantitative PCR (Supplementary Fig. 6). Somatic amplification of the Eµ-*Myc* transgene and the proximal chr19 amplified region as shown for lymphoma #88 (Fig. 1b), was evident in 8/22 tumours. We could also detect recurrent amplification and deletion events involving known human cancer genes[30] (Supplementary Data 3 and Supplementary Fig. 7). Most of these SCNAs were characterized by large segmental amplifications or whole chromosome duplications involving chr3 (including *Nras*), chr6 (including *Kras*), chr10, chr12, chr15 (including *Myc*) and chr18 and monosomy or deletion of part of chr13. Focal SCNAs included low-level gains on chr1 (*Cxcr7*) and chr7 (*Fgfr2*), and hemizygous loss on chr4 (*Psip1*), chr9 (*Nckipsd*) and chr11 (*Msi2*). Taken together these studies highlight the genetic diversity of individual Eµ-*Myc* lymphomas and identify *Bcor* as the most frequently mutated gene in this model of *Myc*-driven malignancy.

**Tertiary driver mutations occur in Eµ-*Myc* lymphomas.** A long-held paradigm established from previous studies using the Eµ-*Myc* model was that a single co-operating lesion in the

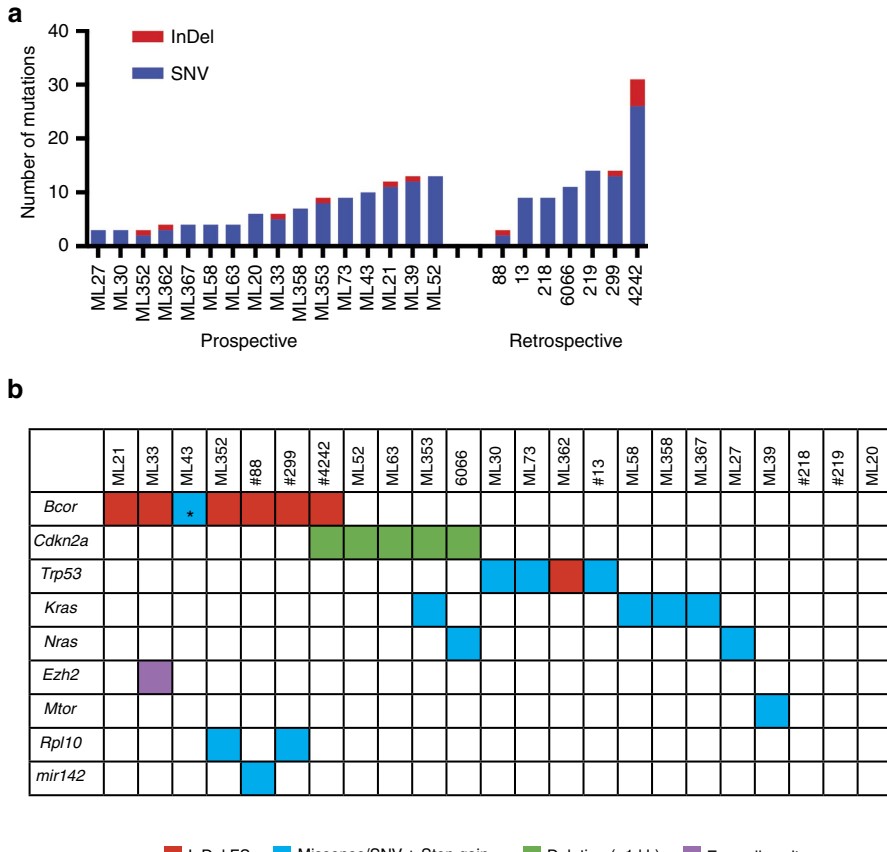

**Figure 2 | Somatic mutations detected by exome-sequencing of Eμ-Myc lymphomas.** (**a**) Mutation frequency across 23 Eμ-Myc lymphomas from a prospective (n = 16) and retrospective series (n = 7). (**b**) Mutations identified in known cancer genes in Eμ-Myc lymphomas.

p19ARF-MDM2-p53 pathway or the RAS pathway would be sufficient to drive tumour development[12,13,31]. However, our data challenges this hypothesis, at least with regard to the role of *Cdkn2a*, as more than one putative 'driver lesion' could be detected in some lymphomas. Examples include: (A) clone #6066 that harboured a *Cdkn2a* homozygous deletion and an activating *Nras*[Q61K] mutation; (B) clone ML353 that carried a *Cdkn2a* homozygous deletion along with an activating *Kras*[Q61H] point mutation and (C) clone #4242 that had a *Cdkn2a* deletion and *Bcor* frameshift deletion. These co-occurring lesions had similar VAF in the individual samples that were validated by deep targeted amplicon sequencing, suggesting they are likely to be present in the dominant tumour clone rather than subclones (Fig. 3). To validate the co-occurrence of multiple driver mutations we transduced cells from two cultured Eμ-Myc lymphomas (#6066 and #4242) with a retroviral barcode, sorted single cells and expanded each clone *in vitro*. DNA sequencing and qPCR CNV analysis of individual clones confirmed a single barcode integration and coalescing lesions in *Cdkn2a* and *Bcor* in lymphoma #4242, and *Cdkn2a* and *Nras* lesions in lymphoma #6066, confirming that both genetic lesions detected in the two lymphomas must be present in the same cell (Supplementary Fig. 8).

A surprising finding from our sequencing of Eμ-Myc lymphomas was the identification of additional putative oncogenic lesions in the context of *Cdkn2a* loss, namely for lymphomas #4242 (*Bcor* frameshift deletion), ML353 (*Kras*[Q61H]) and #6066 (*Nras*[Q61K]). To further determine if Eμ-Myc lymphomas with knockout of one *Cdkn2a* allele not only lost the second allele but also acquired additional mutations we

crossed Eμ-Myc to *Cdkn2a* knockout mice[11] and sequenced spontaneous Eμ-Myc;*Cdkn2a*[+/−] lymphomas. As expected Eμ-Myc;*Cdkn2a*[+/−] mice demonstrated accelerated lymphomagenesis compared with Eμ-Myc transgenic animals (Supplementary Fig. 9). Further WES analysis of six Eμ-Myc; *Cdkn2a*[+/−] lymphomas revealed that all had lost the wild type *Cdkn2a* allele and one lymphoma harboured a heterozygous pathogenic *Kras*[Q61H] mutation (VAF = 67%), indicating selective outgrowth of an Eμ-Myc lymphoma with loss of INK4A/ARF and expression of oncogenic *Kras* (Supplementary Fig. 10 and Supplementary Data 4). Collectively, our data suggest that while heterozygous loss of *Cdkn2a* creates a selective pressure resulting in loss of the wild type *Cdkn2a* allele, it does not ameliorate selective pressure for gain- or loss-of-function of other genes outside the p53-axis. These data are consistent with a previous report indicating that p19ARF inactivation is itself insufficient to trigger lymphomagenesis in Eμ-Myc transgenic mice[32] and indeed the authors of that study posited that 'cryptic mutations other than p19ARF loss accompany the conversion of premalignant Eμ-Myc B cells' in the Eμ-Myc;*Cdkn2a*[+/−] setting. Herein we identify activating mutations in *Ras* and loss of function mutations in *Bcor* as two such events.

**Validating the tumour-suppressor function of *Bcor*.** Protein truncating *BCOR* mutations have previously been reported in both solid and liquid human neoplasms[33–35]. The recurrence of inactivating *BCOR* mutations in human tumours, a recent *in vitro* study showing that BCOR regulates myeloid cell proliferation and differentiation[36] and our observations in the B-cell lineage

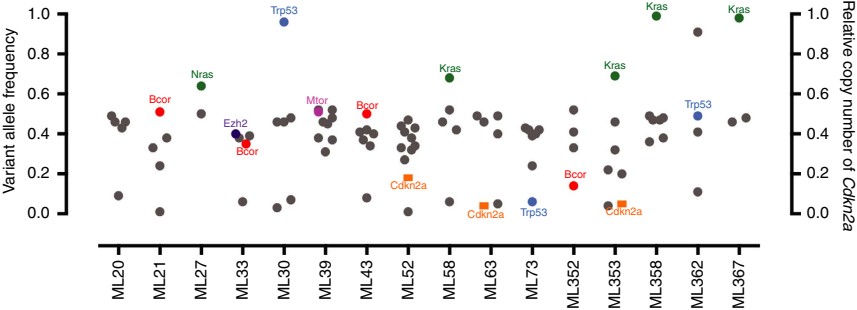

**Figure 3 | Variant allele frequency determined by targeted amplicon sequencing.** Targeted amplicon sequencing (TAM-seq) was performed on the prospective Eμ-*Myc* tumour cohort to validate mutations from WES screen. Variant allele frequency is shown on the left *y*-axis for the mutations that were validated, with genes present in the cancer gene census list highlighted. The right *y*-axis displays relative copy number of *Cdkn2a* in the samples where gene deletion is suspected based on WES read depth and qPCR analysis.

described herein would strongly support a tumour-suppressive function of this gene; however this has not been formally demonstrated through functional *in vivo* studies. To experimentally validate the tumour-suppressor function of *Bcor* in Eμ-*Myc* lymphoma we knocked down or deleted *Bcor* using shRNA or CRISPR-Cas9 methods, respectively, in Eμ-*Myc* haematopoietic progenitor cells and transplanted them into lethally irradiated syngeneic mice (Fig. 4a). We knocked down or deleted *Trp53* as a reference for expected accelerated lymphomagenesis as well as including a negative control (scrambled) shRNA and CRISPR sgRNA sequences. As expected, Eμ-*Myc* lymphomagenesis was significantly accelerated when p53 was knocked down or deleted (Fig. 4b,c). Strikingly, depletion or deletion of *Bcor* using similar techniques significantly accelerated lymphomagenesis demonstrating that Bcor restrains Myc-induced lymphomagenesis (Fig. 4b,c). Western blot analysis of *Bcor* depleted/deleted lymphomas confirmed the loss of Bcor protein expression (Fig. 4d,e). Furthermore, RNA-seq analysis of Eμ-*Myc* lymphomas with CRISPR-Cas9-mediated targeted deletion of *Bcor* showed deletions or insertions proximal to the expected CRISPR-Cas9 trigger sequence in exon 4 that would be expected to cause Bcor loss of function (Supplementary Fig. 11). These experiments demonstrate, for the first time, that *Bcor* can function *in vivo* as a tumour suppressor gene, and plays an important role in Myc-driven lymphomagenesis. To complement these studies, we rescued loss of *Bcor* in the *Bcor*-mutant #4242 Eμ-*Myc* lymphoma. We generated a traceable system introducing wild type BCOR co-expressed with GFP by retroviral transduction. Longitudinal proliferation studies *in vitro* demonstrated that #4242 Eμ-*Myc* lymphomas ectopically expressing wild type *Bcor* were negatively selected in a competitive proliferation assay while expression of GFP alone had no effect (Fig. 5a,b). This finding was confirmed in a similar experiment using the human Namalwa Burkitt lymphoma cell line that expressed very low endogenous levels of *BCOR* (Fig. 5c,d). Our novel functional data using *Bcor* depletion/deletion unequivocally demonstrate that mutation of *Bcor*, seen at a higher frequency in in Eμ-*Myc* lymphomas than lesions in the known cancer genes *Trp53*, *Cdkn2a* and *Ras*, is a 'driver' oncogenic event and provides mechanistic context to observations by others that mutations in *BCOR* occur in a range of human malignancies.

**Unique gene expression signature in *Bcor*^Mut^ lymphomas.**
BCOR has not previously been demonstrated to control tumour cell proliferation or survival, so how BCOR loss co-operates with MYC to drive oncogenesis remains unclear. We found that there was significant enrichment of IgM⁻/IgD⁻ *Bcor*^Mut^ lymphomas

in our tumour cohort (Supplementary Fig. 12). Nine lymphomas with *BCOR* loss-of-function were IgM⁻/IgD⁻, while only one was IgM⁺/IgD⁻. Eight *Bcor*^WT^ tumours were IgM⁻/IgD⁻, while 10 were IgM⁺/IgD⁻. We therefore conclude that *Bcor* mutations are associated with an IgM⁻/IgD⁻ profile (Chi-squared test, $P < 0.05$). To begin to elucidate the underlying biology of the *Bcor*-mutant lymphomas we applied RNA-seq analysis of Eμ-*Myc* lymphomas with shRNA-mediated knock-down of *Bcor* (Eμ-*Myc;shBcor*, $n = 6$) or *Trp53* (Eμ-*Myc;shp53*, $n = 6$); or Eμ-*Myc* lymphomas overexpressing *Nras*^Q61K^ (Eμ-*Myc;Nras*^Q61K^, $n = 5$). Gene expression profiling identified 393 significantly differentially expressed genes in Eμ-*Myc;shBcor* lymphomas compared with those with knockdown of *p53* or expressing oncogenic *Nras* (Supplementary Data 5). To determine if Eμ-*Myc* lymphomas with CRISPR/CAS9-mediated knockout of *Bcor* (Eμ-*Myc;sgBcor*) or those harbouring spontaneous mutations in *Bcor* carried similar gene expression patterns, we clustered gene expression data using the Eμ-*Myc;shBcor* signature gene set (Fig. 6). All sporadic *Bcor* mutant lymphomas and Eμ-*Myc;sgBcor* lymphomas clustered with the Eμ-*Myc;shBcor* lymphomas. This demonstrates that BCOR loss-of-function, either engineered or through spontaneous mutation, drives a unique and reproducible transcriptional pattern in Eμ-*Myc* lymphomas.

To determine if the BCOR loss-of-function gene expression signature provides any molecular insight into the functional interaction between MYC and BCOR we performed pathway analysis using PANTHER. This identified upregulation of TGFβ signalling as the most affected molecular pathway in Eμ-*Myc;shBcor* lymphomas (Bonferroni corrected $P = 0.0058$), with enhanced expression of TGFβ pathway members (*Cited1, Bambi, Acvr2b, Smad3, Mapk12, Tgfb2*). The importance of TGFβ signalling in Eμ-*Myc* lymphomagenesis has been previously reported, where this cytokine derived from macrophages in the tumour microenvironment augmented oncogene-induced senescence[19]. There it was shown that knockout of the senescence-related histone methyltransferase *Suv39h1* accelerated Myc-driven lymphomagenesis with those lymphomas possessing an enhanced TGFβ signature reminiscent of the signature observed in Eμ-*Myc* lymphomas with loss of Bcor. It was proposed that TGFβ drives Myc-induced senescence in a Suv39h1-dependent manner raising the intriguing possibility that BCOR also regulates this unique tumour suppressive mechanism.

**Discussion**
In summary, we have performed the first comprehensive genome-wide sequencing analysis of lymphoid malignancies arising in the Eμ-*Myc* transgenic mouse model. Despite extensive use of the model for cancer gene discovery through forward genetic

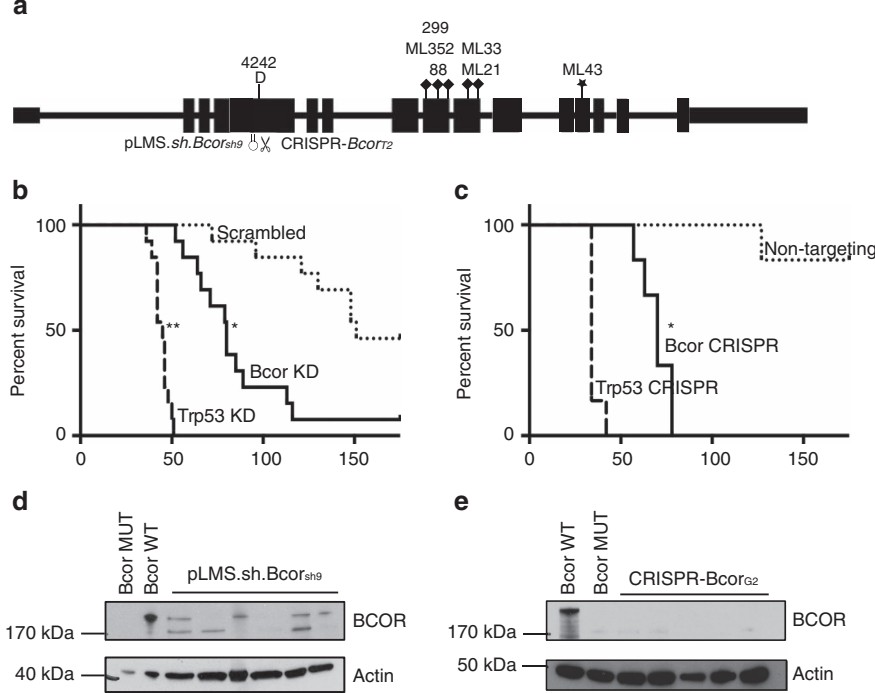

**Figure 4 | Experimental validation of BCOR tumour-suppressor function using RNAi and CRISPR-Cas9.** (**a**) Schematic showing exon structure of *Bcor* with position of identified spontaneous mutations and deletions and regions targeted using shRNA (pLMS-*Bcor.sh9*) or CRISPR-Cas9 (*pCIG-Bcor*$_{G2}$) guide RNA in exome 4. (**b**) Kaplan–Meier curve showing mice injected with $10^6$ GFP + ve Eμ-*Myc* fetal liver cells (FLC) endowed with either pLMS.sh.*Bcor*$_{sh9}$ (solid line), pLMS.*shTrp53*$_{1224}$ (heavy dashed line) or pLMS.sh.Scram (light dashed line). Mice transplanted with Eμ-*Myc* FLC endowed with pLMS.sh.*Bcor*$_{sh9}$ (80 day median survival time post-transplant) or pLMS.*shTrp53*$_{1224}$ (45 day median survival time post-transplant) showed significantly accelerated lymphomagenesis compared with the cohort that received FLC transduced with pLMS.sh.Scram (151 day median survival time post-transplant).
* = P value < 0.05 log-rank (mantel-cox) test, $n = 13$ for each cohort. (**c**) Kaplan–Meier curve showing mice transplanted with $10^6$ Eμ-*Myc* FLC endowed with either CRISPR-*Bcor*$_{G2}$ (solid line), CRISPR-*Trp53* (heavy dashed line) or CRISPR-Scram (lightly dashed line). Mice transplanted with Eμ-*Myc* FLC endowed with CRISPR-*Bcor*$_{G2}$ (70 day median survival time post-transplant) or CRISPR-*Trp53* (34 day median survival time post-transplant) showed significantly accelerated lymphomagenesis compared with the cohort that received FLC transduced with CRISPR-Scram. * = P value < 0.05 log-rank (mantel-cox test), $n = 6$ for each cohort. (**d**) Immunoblot showing levels of BCOR knockdown in six tumours obtained from mouse recipients of Eμ-*Myc* pLMS.sh*Bcor*.$_{sh9}$, #4242 (*Bcor* mutant) and #6066 (*Bcor* wild type). Whole cell lysates were prepared from Eμ-*Myc-Bcor*.sh9 fetal liver-derived tumours, a *Bcor* mutant Eμ-*Myc* lymphoma cell line (#4242) and a *Bcor* WT Eμ-*Myc* lymphoma cell line (6066). Western blot analysis was performed with antibodies specific to BCOR. Reduced BCOR protein expression was demonstrated in the Eμ-*Myc* pLMS.sh*Bcor*.$_{sh9}$ lysates compared with WT. Equivalent protein loading was confirmed by probing for β-Actin. (**e**) Whole cell lysates were prepared from Eμ-*Myc*-pCIG-Bcor$_{G2}$ fetal liver-derived tumours, a *Bcor* mutant Eμ-*Myc* lymphoma cell line (4242) and a *Bcor* WT Eμ-*Myc* lymphoma cell line (6066). Western blot analysis was performed as in **d** above. No BCOR protein expression was apparent in the Eμ-*Myc*-pCIG-Bcor$_{G2}$ lysates compared with WT.

approaches, we have discovered new co-operative genetic events leading to spontaneous B-cell lymphomas. Our discovery of the co-occurrence of multiple known driver mutations within a single Eμ-*Myc* lymphoma suggests that individual clones evolve over time, requiring multiple co-operative events to enable malignant transformation or select for a more aggressive clone. Whether the order of mutational events is important is not clear. However, heterozygous *Cdkn2a* deletions do not completely ameliorate the selective pressure for acquisition of mutations in other cancer genes outside the p53-axis (for example, *Nras*, *Kras*, *Bcor*). We identified a high-frequency of somatic *Bcor* mutations in Eμ-*Myc* lymphomas and subsequently have shown that genetic disruption of *Bcor* can accelerate Eμ-*Myc* lymphomagenesis. Among human haematological cancers, *BCOR* mutations have been reported in an array of cancer types including acute myeloid leukaemia and myelodysplastic syndromes, chronic lymphocytic leukaemia and acute lymphoblastic leukaemia (Supplementary Data 6). We have shown experimentally, for the first time, that *Bcor* can act as a tumour suppressor gene and the distinct gene-expression signature of *Bcor*-mutant lymphomas suggests that BCOR loss-of-function may subvert TGFβ signalling to drive lymphoma

development. Collectively our studies provide important information on the genomic architecture of one of the most utilized GEMMs, identifies *Bcor* as a *bona fide* tumour suppressor gene and provides important information regarding the functional interaction between Myc and p19ARF in the context of Eμ-*Myc* lymphomagenesis.

## Methods

**Eμ-*Myc* Lymphomas.** All animal work was performed with approval from the Peter MacCallum Cancer Centre Animal Experimentation Ethics Committee. A prospective series of sixteen heterozygote Eμ-*Myc* mice originally obtained from the Walter and Eliza Hall Institute (Melbourne, Australia) on a C57BL/6 background were bred for discovery analysis and a retrospective series of seven lymphomas were collated from archived laboratory resources. The age and gender of mice used in the study are detailed in Supplementary Table 1. Matched normal tail tissue was available for lymphoma #88 and a littermate control ML62. We collected peripheral blood for isolation of B-cells from 4-weeks of age and at 2-week intervals until the mice were sacrificed with advanced disease.

**Nucleic acid extraction.** DNA was extracted using the DNeasy blood and tissue kit (Qiagen, Hilden, Germany), RNA was extracted using the miRNeasy mini kit (Qiagen) and nucleic acids were quantified by spectrophotometer. RNA integrity was assessed using the Agilent Bioanalyzer 2100 (Agilent, CA, USA).

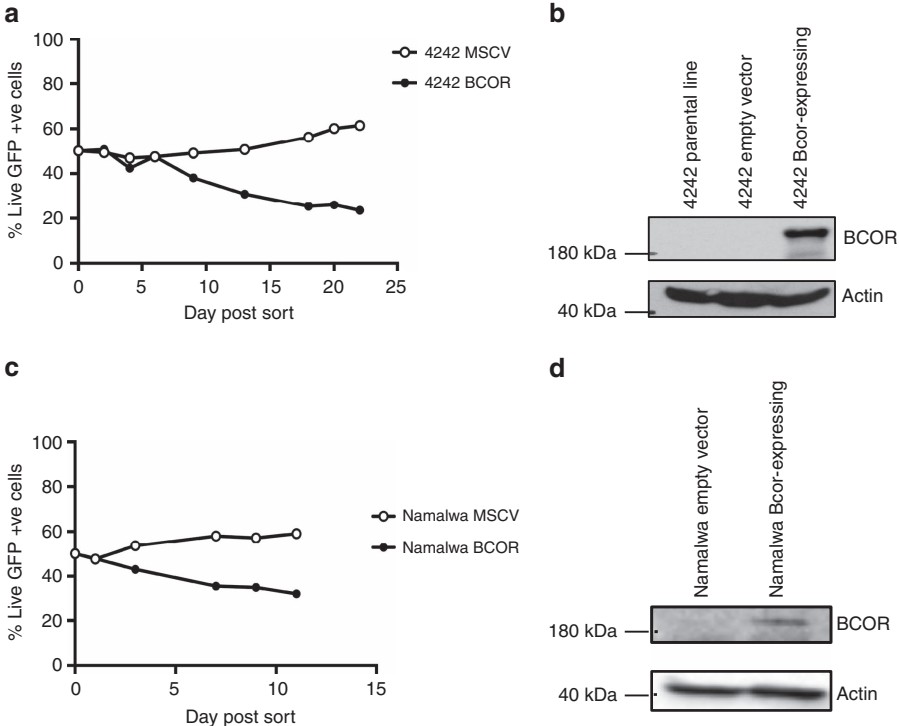

**Figure 5 | *Bcor* re-expression in a *Bcor* null Eµ-*Myc* cell line. (a)** 4242 cells (*Bcor* null) were transduced with either MSCV inert vector or MSCV;*Bcor*[wt] and FACS sorted into a ratio of 50:50 with the non-transduced parental cell line. The GFP population was monitored over time and demonstrated that forced *Bcor* expression in 4,242 cells is a competitive disadvantage. **(b)** Western blot analysis showing BCOR re-expression in the 4,242 cell line that was transduced with MSCV;*Bcor*[wt] compared with the cells transduced with the inert vector and the parental cell line. **(c)** Human B-cell line, Namalwa (Bcor-low) were transduced with either MSCV inert vector or MSCV;*Bcor*[wt] and FACS sorted into a ratio of 50:50 with the non-transduced parental cell line. **(d)** Western blot analysis showing BCOR re-expression in the Namalwa cell line that was transduced with MSCV;*Bcor*[wt] compared with the cells transduced with the inert vector.

**Whole-genome sequencing.** Approximately 1 µg of gDNA was prepared for fragment-sequencing libraries and then processed according to standard protocols using TruSeq chemistry (Illumina, San Diego, CA, USA). Mate-pair libraries were prepared using Nextera Mate Pair Sample Prep Kit according to manufacturers instructions (Illumina, San Diego, CA, USA). Paired-end sequencing (2 × 100 bp) on HiSeq2000 (Illumina) was applied to achieve 1-30-fold mapped read coverage across the entire genome depending on required depth (Supplementary Data 7). Reads were aligned using Bowtie2 v2.1.0 (Langmead and Salzberg 2012) to mouse reference genome mm10 (Dec. 2011 GRCm38/mm10). Duplicates were identified and removed from the aligned data using Picard's Mark Duplicates v1.89 (Broad Institute, Boston, MA, USA). BreakDancer v1.3.5 (ref. 37), CREST v0.0.1 (ref. 38) and Socrates v0.9.5 (ref. 39) were employed to identify structural variants. Whole genome *de novo* assembly was undertaken using Gossamer[40] to compare resulting contigs with the genomic sequence relevant to the transgene. Control-FREEC (ref. 41) was used to call somatic copy-number alterations using tail88 as the germline control. A window size of 50,000 bp was used.

**Exome sequencing.** Approximately 1 µg of gDNA prepared as above was processed for whole exome-capture resequencing according to standard protocols (Agilent Technologies, Santa Clara, CA, USA) on a HiSeq 2000 achieving a mean sequence depth of 110-fold and at least 10-fold coverage in 98% of targeted bases (Supplementary Data 8). Variant calling methods were based on Genome Analysis Tool Kit (GATK)[42] muTect[43] and Strelka[44]. An intersection of variant callers was used to optimize specificity and sensitivity by including only SNVs called by muTect and one other method or any InDel called by at least two methods. Variants were annotated using the ENSEMBL database[45]. Somatic copy-number calls from exome data were generated using ADTEx (ref. 46) against three independent lymphoma samples.

**RNA sequencing and data analysis.** Approximately 1 µg of RNA was used to generate polyA enriched cDNA libraries using TruSeq sample preparation kit (Illumina, San Diego, CA, USA) and paired-end RNA-sequencing (2 × 50 bp) was performed on the HiSeq 2000, generating 4 × 10[7] reads per sample. Reads were quality checked using FastQC and trimmed if necessary for low base quality or

adaptor using Cutadapt v1.6 (ref. 47). RNA-sequencing reads were aligned using the short read aligner software – TopHat[48]. The raw RNA-sequencing data was converted to feature counts using HTSeq software package allowing generation of expression matrices collating read counts per gene based on ENSEMBL annotation[45]. EdgeR v3.0 (Bioconductor, Fred Hutchinson Cancer Research Center, Seattle, WA, USA) statistical software was used in the statistical programming tool R (http://www.r-project.org/) to normalize gene expression and perform statistical analysis by linear regression[49]. Gene Cluster 3.0 was used to cluster samples and genes by average linkage centred clustering. Cluster diagrams and heat maps were visualized and then exported from Treeview v1.1.6r2. Pathway mapping of significant genes (P value < 0.05) was undertaken using PANTHER pathway analysis software[50].

**Variant validation by amplicon-based gene sequencing.** Targeted amplicon-based massively-parallel sequencing was used to validate mutations detected by exome-sequencing and for analysis of serial blood samples. PCR Primers were designed flanking mutated bases incorporating a 5-prime extended universal sequence for secondary PCR enabling addition of full-length Illumina adaptors (Supplementary Data 9). Primary PCR reaction mixture included 10 ng genomic DNA template, 10 µl of 2 × Phusion high-fidelity PCR master-mix (New England Biolabs, Ipswich, UK), 2.5 µl of 4 µM forward and reverse oligonucleotide primers and made up to contain a total reaction volume of 20 µl with PCR-grade $H_2O$. The PCR reaction mixture was cycled at 95 °C for 2 min, (95 °C for 30 s 68 °C (and decreasing 1 °C every cycle) for 30 s, 72 °C for 30 s) for 9 cycles, (95 °C for 30 s, 61 °C for 30 s, 72 °C for 30 s) for 32 cycles and 72 °C for 5 min. PCR products were then pooled and combined with AMPure XP magnetic beads (Beckman Coulter, Pasadena, CA, USA) in a ratio of 1:0.9 and then purified as per manufacturer protocol. Purified pooled amplicons were diluted 1:100 in PCR-grade $H_2O$ and prepared in PCR reaction mix of 1 µl template DNA, 10 µl of 2 × Phusion high-fidelity PCR master-mix, 2.5 µl of 4 µM forward and reverse oligonucleotide primers and made up to contain a total reaction volume of 20 µl with PCR-grade $H_2O$. The PCR-reaction mix was then subject to an additional PCR cycle using indexed Fluidigm CS1 and CS2 primers (Fluidigm, CA, USA) at conditions of 95 °C for 5 min, (95 °C for 15 s, 60 °C for 30 s, 72 °C for 1 min) for 15 cycles and 72 °C for 3 min. Secondary PCR products were then purified using AMPure XP

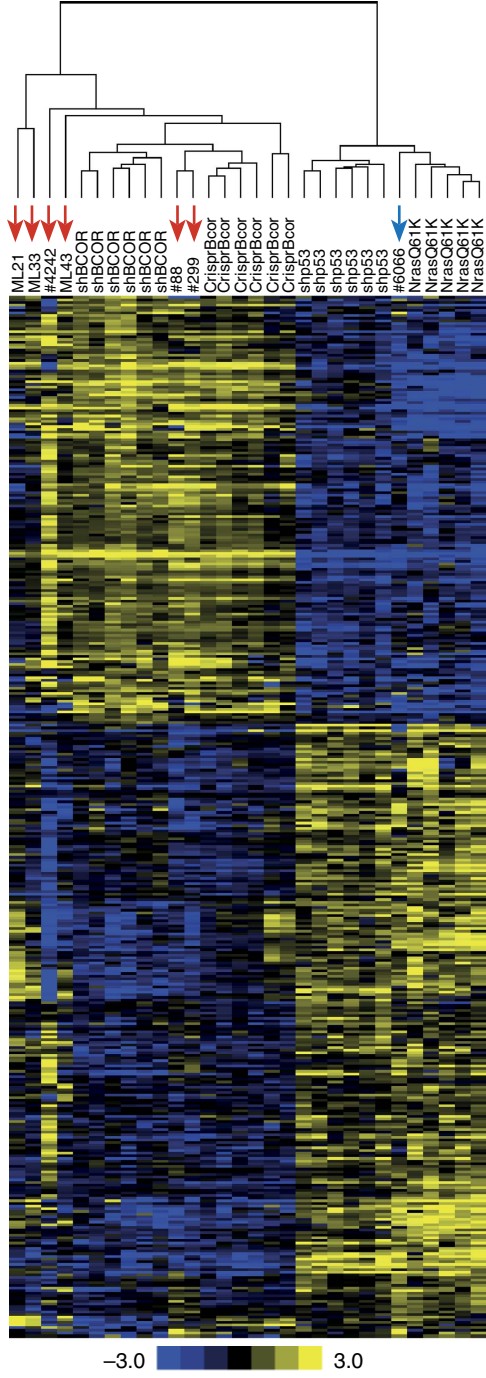

**Figure 6 | Gene-expression profiling of Eμ-Myc lymphomas identified a reproducible signature of Bcor mutation or knockdown.** RNA-seq analysis was first used to identify 393 significantly differentially expressed genes (FDR-corrected $< 0.05$ and $\log_2$ fold-change $\geq 2$) between pLMS.sh.$Bcor_{sh9}$ and combined pLMS.sh.$Bcor_{sh9}$ and overexpressing $Nras^{Q61K}$ fetal liver-derived Eμ-Myc lymphomas. RNA-seq data for shRNA fetal liver-derived lymphomas plus CRISPR-Cas9 (CrispR.Bcor$_{T2}$) and sporadic Eμ-Myc lymphomas was then clustered using the 393 gene set. All CrispR.Bcor$_{T2}$ and sporadic Bcor mutant lymphomas (red arrows) cluster with the pLMS.sh.$Bcor_{sh9}$ fetal liver-derived lymphomas. The sporadic $Nras^{Q61K}$ mutant cell line #6066 (blue arrow) also clustered correctly with $Nras^{Q61K}$ overexpressing fetal liver-derived lymphomas. Heatmap and scale bar represents median normalized log2-fold gene-expression.

magnetic beads as previously described. Purified DNA was quantified and loaded on MiSeq ($2 \times 150$ bp) (Illumina, CA, USA) at 6 ρM and produced a yield of $\sim 4.5$–5 Gb. Custom ports on the MiSeq cartridge were used to incorporate the custom sequence tagged (CS) Fluidigm primers CS1 (5′-ACACTGACGACATGG TTCTACA-3′) (port 18, read 1), CS2RC (5′-AGACCAAGTCTCTGCTACC GTA-3′) (port 19, index read) and CS2 (5′-TACGGTAGCAGAGACTTGGTCT-3′) (port 20, read 2), which were used at a final concentration of 0.5 μM. Amplicon sequencing data were aligned using bwa-mem (v0.7.12) to the mouse reference genome (GRCm38). A list of regions of interest (ROIs) was compiled as a union set of all expected variants in the EuMyc mice. Pileup data were generated using samtools (v1.1) over the ROIs. Read depth and the number of reads supporting each of the expected variants were extracted from pileup data. Varscan (v2.3) was used for variant calling to identify potential variants outside the ROIs.

**Cellular barcoding.** A degenerate 16 nucleotide sequence was designed to theoretically provide 4,194,304 unique barcodes for cell tracking. A PhiX DNA spacer sequence (138 bp) plus flanking 5′ and 3′-prime universal priming sites (CS1 and CS2, 22 bp each) were incorporated to enable PCR amplification and direct DNA sequencing of barcodes. PCR primers designed complimentary to CS1 and CS2 and harbouring 5′-prime XhoI and EcoRI restriction sites enabled sequential PCR amplification and then digestion of the double stranded DNA barcoded fragment for directional shotgun cloning into an MSCV-IRES-BFP retroviral plasmid. A complex pool of recombinant MSCV-Barcode plasmid was used to generate retrovirus in HEK293T cells in combination with the pAMPHO viral packaging vector (ClonTech). Eμ-Myc Cell lines #6066 and #4242 were transduced at 0.2 multiplicity of infection. Transduced cells were single cell sorted by BFP expression and deposited into single wells of a 96-well plate containing culture media using a BD FACSAria Fusion (BD Biosciences). Eμ-Myc clones were grown by cell culture expanding to $\sim 10^6$ cells followed by genomic DNA extraction by column chromatography (Qiagen). Barcode sequences were amplified from genomic DNA by PCR using Phusion master mix (NEB) with CS1 and CS2 primers using 28-cycle reaction (Appendix 3B). Barcodes were then sequenced by Sanger sequencing from CS2 primer.

**Western blot antibodies.** Western blot analysis was performed using affinity-purified rabbit-α-Bcor[51] used at 1:3,000, mouse-α-phospho-p44/42 MAPK (Erk1/2) clone E10 #9106 (Cell Signaling Technology) used at 1:1,000, rabbit–α-phospho-STAT5 (Tyr694) clone D47E7 #4322 (Cell Signaling Technology), used at 1:1,000, rabbit–α-STAT5 #9363 (Cell Signaling Technology) used at 1:1,000, rabbit–α-phospho-JAK2 (Tyr1007/1008) #3771 (Cell Signaling Technology), used at 1:1,000, rabbit-α-JAK2 clone D2E12 #3230 (Cell Signaling Technology), used at 1:1,000, rabbit–α-phospho-S6RP (Ser240/244) #2215 (Cell Signaling Technology) used at 1:1,000, mouse-α-αtubulin clone DM1A #MABT205 (Merck Millipore) used at 1:5,000, rabbit-α-HSP90 #4874 (Cell Signaling Technology) used at 1:5,000 or mouse-α-βActin clone AC-74 #A2228 (Sigma Aldrich) used at 1:5000. Secondary antibodies used were polyclonal rabbit-anti-mouse-HRP #P026002-2 (Dako) used at 1:10,000 and polyclonal swine-anti-rabbit-HRP #P039901-2 (Dako) used at 1:10,000.

Original western blots that correspond to blots shown in Figs 4 and 5 are shown in Supplementary Figs 13 and 14.

**Viral transduction.** Calcium phosphate transfections of packaging cell lines were used to generate retroviral supernatant for vectors pLMS-Bcor.sh9, pLMS-Trp53.sh1224, pLMS-Scram, MSCV-Nras.Q61K, MSCV-empty, MSCV-Bcor, pCIG-BcorG2, pCIG-Trp53(b) or pCIG-Scram (Supplementary Table 2). Eμ-Myc fetal cells were incubated with the retroviral supernatant and appropriate media to facilitate retroviral transduction with GFP positivity of the cells used as marker of transduction efficiency. Irradiated recipient C57BL/6 mice received $10^6$ transduced GFP expressing cells via intravenous injection[37] and analysis of tumour latency was performed. Cell lines #4242 and Namalwa were incubated with retroviral supernatant containing MSCV-empty or MSCV-Bcor. Cells transduced with MSCV-Bcor or MSCV-empty vector were sorted and combined with non-transduced parental lines in a 50:50 ratio of GFP-positive to GFP-negative cells. Human cell lines were sourced from the American Type Culture Collection (ATCC).

**Data availability.** The datasets generated during and/or analysed during the current study have been deposited in the NCBI SRA (http://www.ncbi.nlm.nih.gov/sra) database under accession code SUB2009648. The authors declare that all data supporting the findings of this study are available within the article and its supplementary information files or from the corresponding author on reasonable request.

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

## Acknowledgements

We thank Charles Mullighan and Ari Melnick for helpful discussion and Jerry Pelletier for providing us with the pCIG-constructs. We thank members of the Peter Mac Molecular Genomics Core Facility for their contribution. We acknowledge the following funding agencies: Leukaemia Foundation of Australia, Arrow Bone Marrow Transplant Foundation, National Health and Medical Research Council Australia, Cancer Council Victoria, Victorian Cancer Agency, Australian Cancer Research Foundation, Peter MacCallum Cancer Centre Foundation, National Institutes of Health.

## Author contributions

Experiments were performed by M.L., R.W.T., E.K., E.D.H., G.P.G., B.P.M., M.J.K., I.T., M.W. Experiments were designed by. M.L., R.W.T., E.K., E.D.H., J.S., G.M.M., L.K., M.D.G., V.J.B., R.D.H., A.T.P. and R.W.J. Data were analysed by M.L., R.W.T., E.K., E.H., J.S., G.M.M., M.A.D., R.L., J.L., J. Sch., L.K., A.T.P. and R.W.J. Reagents were provided by S.C., G.P., D.C., M.B., M.D.G., V.J.B., R.A.D. The manuscript was written by M.L., R.W.T., E.K., E.D.H., J.S., G.M.M., L.K., M.D.G., V.J.B., R.A.D., R.D.H., A.T.P. and R.W.J.

## Additional information

**Competing financial interests:** The authors declare no competing financial interest.

