## [Peer Review File · Nature Communications]

Reviewer #1 (Remarks to the Author)

In this manuscript by Lefevure et al. entitled "Genomic characterisation of Eμ-Myc lymphomas by massively-parallel sequencing identifies Bcor as a MYC co-operative tumor-suppressor gene", the authors sequence spontaneously arising lymphomas arising in the Eu-Myc model. This model has been quite powerful as a cancer gene-discovery tool and for linking therapy response to genotype, a feature that makes the current study important and of significant interest to a large audience. The experiments appear well performed and in general support the conclusions. A few comments for improvement/clarification.

- 1) How were the sequenced lymphoma tested for clonality (or were they)? What's the percentage of contaminating/normal sample? (was VDJ/IgH status tested)? I see the "a posteriori" experiment with barcoding which supports the lesions are occurring in the same cell, but it would be nice to have a measure of what has been covered in terms of cell population.
- 2) mTOR mutation. Is this mutation at a conserved residue and has it already been reported in publicly available studies, for example Sanger Database, CCLE, etc?
- 3) Lesion association. Are any of the co-occurring lesions in figure 2 statistically significant?
- 4) JAK2 gain. The authors note the presence of JAK2 amplification and test its oncogenic effect by forced expression of JAK2 activating mutation. Observing no acceleration, they conclude JAK2 has no oncogenic role in Eμ-myc progression. However, overexpression of the mutant in a context where the gain is already present (as JAK2 dosage is already increased in Eμ-myc mice) might be not the right setup to test the oncogenic impact of such lesion. To their credit they use ruxolitinib to inhibit JAK2 and also document no effect (Fig S4). Can the authors strengthen this by providing evidence that ruxolitinib was working in their lymphomas to block signaling? Alternatively, they could suppress JAK2 expression and assess the consequences.
- 5) The authors should comment on why the CRISPR scramble shows much lower background in lymphoma onset then the Scram shRNAs (Fig. 3).
- 6) BCOR. The authors are probably aware BCL6 can have tumor suppressive functions in pre-B cells (differently from germinal center derived lymphomas). Do they see (inverse) correlation with BCL6-based signatures by using PANTHER, GSEA or similar approaches? Do they observe a shift of the BCOR-deleted lymphoma toward the pre-B phenotype?
- 7) Is there association of MYC amplification/translocation and BCOR loss/mutation in human Burkitt or B-ALL?

Reviewer #2 (Remarks to the Author)

In this paper Lefebure et al., use a simple and thorough strategy to identify co-operative mutations that promote MYC-driven lymphomagenesis in the classic Eu-MYC lymphoma model. They further validate their findings by loss of function experiments of BCOR, the top candidate gene mutated in their samples. Overall, this study has been well-performed. I therefore recommend this study for publication after few minor changes.

My suggested changes are as follows:

- (1) In order to filter polymorphisms from their WGS data they use normal tail DNA from two animals. I suggest this subset be expanded to include either more tail DNA from Eu-lymphoma mice or B-cells from normal mouse strain corresponding to the Eu-MYC lymphoma.
- (2) Although the authors prove elegantly through both shRNA and sgRNA approaches that BCOR acts as a tumor suppressor in Eu-MYC lymphomagenesis, I would prefer they complement these studies through over expression of BCOR in human B-lymphoma lines. Such studies are essential to prove the functional significance of BCOR as a tumor suppressor in human lymphomas.
- (3) BCOR is a binding partner of BCL6. BCL6 is key oncogene in MYC-driven lymphomas for which targeted treatments are currently being developed. Can the authors comment on the interactions

between BCL6 and BCOR in this setting, through DNA binding studies, and BCL6-BCOR interactions in their samples?

Reviewer #3 (Remarks to the Author)

Lefebure et al.,

In their paper (Lefebure et al.,) the authors describe the genome sequencing and analysis of lymphomas from the Eu-Myc lymphoma model, which is very widely used to model human lymphoma development. The paper is well written but I have concerns about the contribution of this work as it is presented, and some of the analysis as detailed below.

1. A single whole genome was generated to map the transgene. Then 23 tumor exomes were sequenced, with just two tail DNA samples used as germline controls. I would say this is far from a comprehensive analysis.

- Firstly, what was the exact phenotype of these tumors? Were they thymic, splenic or did the lymphoma develop in another organ? Why sequence these tumours? What was the sample selection rationale?

- Did they all have the same development/marker expression profile? (rather basic information is provided in supplementary Table 1)

- It is stated that two tails were sequenced for filtering and that the validation rate of the variants was +93%. Was the validation performed on the tumor DNA or using tumor AND matched normal DNA? If it's the former then there is a real potential that germline variants have slipped through the calling. There are several erroneous mouse cancer sequencing studies that have been published where match tumor/germline DNA was not used for calling. DNA "from the same colony" is not an appropriate control.

- For the abovementioned validation - what does a validate rate of 93% actually mean? How were these variants selected?

- The indel to SNV ratio is lower than I would expect, at least based on other cancer genome studies.

2. The list of drivers is descriptive. No statistics are applied to this analysis. It goes without saying that Bcor and Cdkn2a are probably mutated in a significant number of cases but other genes such as Ezh2 and Mtor are discussed but there is no formal proof that these genes are mutated at a frequency greater than expected by chance. Likewise for Rpl10.

3. My next point is on the claims surrounding Bcor. This gene has been described previously as being a driver in ALL, AML, T-cell prolymphocytic leukemia, sarcoma and in various other tumor types. Various mechanisms of these loss-of-function mutations have been described - Most being related to BCORs co-repressor function. The suggestion in this paper (although not proven statistically) is that Myc and Bcor co-operate, and this is shown functionally with shRNA and CRISPR experiments, and by expression profiling to show that Bcor mutant tumors have a different expression signature. This again is descriptive and there is no real attempt to define a mechanism, even using the available expression data, beyond the observation that there is TGFb pathway upregulation. The result is interesting but what does it really mean?

4. My final point is the relevance of the Bcor-Myc observation to human cancer. Is there a statistically significant co-occurrence of these mutations in human cancers, in particular in Myc driven lymphoma? If not how does this speak to the value of the Eu-Myc lymphoma model?

Response to Reviewer comments

Reviewer #1 (Remarks to the Author): Expert in the Eu Myc mouse model

In this manuscript by Lefebure et al. entitled "Genomic characterisation of Eμ-Myc lymphomas by massively-parallel sequencing identifies Bcor as a MYC co-operative tumor-suppressor gene", the authors sequence spontaneously arising lymphomas arising in the Eu-Myc model. This model has been quite powerful as a cancer gene-discovery tool and for linking therapy response to genotype, a feature that makes the current study important and of significant interest to a large audience. The experiments appear well performed and in general support the conclusions. A few comments for improvement/clarification.

Question 1. How were the sequenced lymphoma tested for clonality (or were they)? What's the percentage of contaminating/normal sample? (was VDJ/IgH status tested)? I see the "a posteriori" experiment with barcoding which supports the lesions are occurring in the same cell, but it would be nice to have a measure of what has been covered in terms of cell population.

Response 1. We thank the reviewer for recognizing that our barcoding experiment provides evidence of lymphoma clonality. In addition to those data, the clonality of the Eμ-Myc lymphomas can be inferred from the variant allele frequency (Supplementary Figure S8). Most variants cluster at VAF ~0.5 as would be expected for a monoclonal pure sample where a heterozygous somatic mutation lies in a diploid chromosomal region. Purity of the sample can also be inferred from the CNV profiles including qRT-PCR analysis of *Cdkn2a* also shown in Supplementary Figure S8. There are some examples where Eμ-Myc lymphomas may be polyclonal (e.g. ML73 and ML352) where the driver gene mutations (*Trp53* and *Bcor*, respectively) have VAF<0.5. Clonality can also be inferred from the FACS analysis showing that most cases had homogenous IgM and IgD profiles (see new Supplementary Figure S14 and Updated Supplementary Table 1). Importantly, we posit that even if a small proportion of the cases may be polyclonal, this does not impact the major findings of our study designed primarily as a screen for new driver gene mutations arising spontaneously in Eμ-Myc. As recognized by the reviewer, the co-occurrence of some mutations was also validated by cellular barcoding and single cell cloning, therefore making the point that Eμ-Myc lymphomas can arise from the acquisition of tertiary mutations.

Question 2. mTOR mutation. Is this mutation at a conserved residue and has it already been reported in publicly available studies, for example Sanger Database, CCLE, etc?

Response 2. The *Mtor* mutation identified for ML39 affected a conserved residue and was predicted to be deleterious using the SIFT algorithm. The mutation has not been observed within human cancers (see figure below with the red arrow showing the location of the Eμ-Myc *Mtor* mutation with respect to other *MTOR* mutations within COSMIC v77). We have revised the results section of the text to highlight the point raised by the reviewer.

Question 3. Lesion association. Are any of the co-occurring lesions in figure 2 statistically significant?

Response 3. We screened insufficient number of samples to make a statistically robust statement regarding the co-occurrence of driver gene mutations. We recognize that based on our exciting and novel findings that an expanded study appropriately powered to provide meaningful statistics regarding co-occurring lesions would be of interest however such a large study is beyond the scope of our current project.

Question 4. JAK2 gain. The authors note the presence of JAK2 amplification and test its oncogenic effect by forced expression of JAK2 activating mutation. Observing no acceleration, they conclude JAK2 has no oncogenic role in E μ -Myc progression. However, overexpression of the mutant in a context where the gain is already present (as JAK2 dosage is already increased in E μ -Myc mice) might be not the right setup to test the oncogenic impact of such lesion. To their credit they use ruxolitinib to inhibit JAK2 and also document no effect (Fig S4). Can the authors strengthen this by providing evidence that ruxolitinib was working in their lymphomas to block signaling? Alternatively, they could suppress JAK2 expression and assess the consequences.

Response 4. We have now provided a Western blot (New Supplementary Fig 4B) which shows that pSTAT5^{Tyr694}, a biomarker of activated JAK2 signalling, is undetectable in the E μ -Myc cell line #4242, indicating very low JAK-STAT signaling in these cells. This further supports our belief that JAK2 plays no major role in E μ -Myc lymphomagenesis. In the new Western blot we have provided a control cell line (SET2) containing mutant JAK2 (V617F) where the endogenous level of pSTAT5^{Tyr694} is high but is then significantly reduced in response to ruxolitinib, demonstrating the on target activity of the JAK2 inhibitor in these experiments. We have provided additional text in the results section to highlight these new data.

Question 5. The authors should comment on why the CRISPR scramble shows much lower background in lymphoma onset then the Scram shRNAs (Fig. 3).

Response 5. Variation in disease latency is common between fetal liver experiments using the Eu-Myc model and may be due to a variety of factors including variation

between batches of harvested fetal liver stem cells and efficacy of irradiation of the host (recipient) mice. We are aware that such variation can occur and to address this we always include controls such as CRISPR or shRNA constructs targeting the known co-operative genes (e.g. *trp53*) in each experiment. This allows us to accurately determine if knockdown, knockout or overexpression of a given gene in Eu-Myc fetal liver progenitor cells truly does provide a functional genetic interaction with transgenic Myc and decrease the latency of lymphomagenesis. Examination of the data in Figure 3 shows that the latency for the p53 shRNA and p53 CRISPR experiments are very similar as is the latency for the Bcor shRNA and Bcor CRISPR experiments even though the latency for the SCR shRNA and CRISPR controls experiments are somewhat different. This indicates the power of the system to identify strong genetic interactions between Myc and mutations in cooperating genes such as *p53* and *Bcor*.

Question 6. BCOR. The authors are probably aware BCL6 can have tumor suppressive functions in pre-B cells (differently from germinal center derived lymphomas). Do they see (inverse) correlation with BCL6-based signatures by using PANTHER, GSEA or similar approaches? Do they observe a shift of the BCOR-deleted lymphoma toward the pre-B phenotype?

Answer 6. We thank the reviewer for this question and indeed we did search for any involvement of Bcl6. PANTHER pathway analysis did not find an enrichment of BCL6 target genes between Eμ-Myc;*shBcor* and other FL-derived Eμ-Myc lymphomas. As there are no BCL6 gene sets represented in GSEA MSigDB gene sets we created a custom BCL6-BCOR co-occupancy gene set generated from an intersection of those genes identified by both both BCL6 and BCOR ChIP-seq experiments of Ly6 B-cells, as previously described by Hatzi *et al* (Cell Reports 4(3)p578). GSEA analysis showed no enrichment of the BCL6-BCOR gene set in either direction when contrasting *Bcor*^{Mut} and *Bcor*^{WT} lymphomas. Due to space constraints we were not able to include an extensive discussion on this point however if the Editor and reviewer felt that this was necessary we would be happy to provide a revised discussion as required.

With regard to the immunophenotype of the Eμ-Myc *Bcor*^{Mut} lymphomas we found that there was significant enrichment of IgM⁻/IgD⁻ lymphomas in the BCOR^{Mut} lymphomas (New Supplementary Figure 14). Nine lymphomas with *Bcor* loss of function were IgM⁻/IgD⁻ while only one was IgM⁺/IgD⁻. Eight *Bcor*^{WT} tumours were IgM⁻/IgD⁻ while ten were IgM⁺/IgD⁻. We therefore conclude that Bcor mutations are associated with an IgM⁻/IgD⁻ profile (Chi-squared test, p<0.05). This information has been included in the revised text along with New Supplementary Figure 14.

Question 7. Is there association of MYC amplification/translocation and BCOR loss/mutation in human Burkitt or B-ALL?

Response 7. To our knowledge *BCOR* mutations have not been described in any large-scale sequencing efforts for Burkitts lymphoma and are not represented in the COSMIC database for this disease type. As shown in Supplementary Table 11 *BCOR* mutations arise infrequently in B-ALL and CLL. Due to the low frequency of *BCOR* mutations in these diseases it would not be possible to show any statistical correlation between MYC translocations or amplifications and *BCOR* mutations in these leukemias until more human sequencing data is available.

Reviewer #2 (Remarks to the Author): Expert in lymphoma

In this paper Lefebure et al., use a simple and thorough strategy to identify co-operative mutations that promote MYC-driven lymphomagenesis in the classic Eu-MYC lymphoma model. They further validate their findings by loss of function experiments of BCOR, the top candidate gene mutated in their samples. Overall, this study has been well-performed. I therefore recommend this study for publication after few minor changes.

My suggested changes are as follows:

Question 1. In order to filter polymorphisms from their WGS data they use normal tail DNA from two animals. I suggest this subset be expanded to include either more tail DNA from Eu-lymphoma mice or B-cells from normal mouse strain corresponding to the Eu-MYC lymphoma.

Response 1. We agree that private SNPs in the germline can be mistaken for somatic mutations in the absence of matched normal DNA sample for every animal. To overcome this issue we used a stringent filtering criteria whereby we removed identical SNVs detected in 2 or more lymphomas. These SNVs were carefully curated so as not to remove important recurrent hotspot mutations in oncogenes (e.g. *Nras*). We found no compelling evidence for hotspot mutations outside of the known *Ras* hotspot mutations. Furthermore, SNVs arising in a subset genes of interest (e.g. *Mtor*, *Ezh2*) were validated by targeted sequencing of DNA isolated from the blood of the same animal taken a 4 weeks after birth before clonal expansion (see New Supplementary Figure S5). We clearly show that somatic variants detected at end stage were either undetected or had a very low VAF (e.g. ML39 MTOR) in the blood at 4 weeks indicating that these variants were almost certainly not private heterozygous polymorphic SNPs.

Question 2. Although the authors prove elegantly through both shRNA and sgRNA approaches that BCOR acts as a tumor suppressor in Eμ-MYC lymphomagenesis, I would prefer they complement these studies through over expression of BCOR in human B-lymphoma lines. Such studies are essential to prove the functional significance of BCOR as a tumor suppressor in human lymphomas.

Response 2. We have now conducted additional functional experiments requested by the reviewer involving overexpression of BCOR in the BCOR^{MUT} Eμ-Myc lymphoma #4242 and human lymphoma line Namalwa that we found to intrinsically express very low levels of *BCOR* expression (New Supplementary Figure 13). We transduced each cell line with an MSCV-based retroviral vector to overexpress BCOR and as a control transduced the empty MSCV vector alone. The MSCV retroviral vector we used expresses the GFP reporter protein allowing us to track the proliferation of transduced cells. We used a competition assay to measure the proportion of transduced MSCV BCOR cells in culture over time by comparing cells transduced with MSCV empty vector using GFP detection starting with 50:50 mixture. In both the Eμ-Myc #4242 lymphomas and Namalwa human cell line we observed that expression of BCOR was selected against in these competitive proliferation assays consistent with a tumor suppressor function for BCOR.

Question 3. BCOR is a binding partner of BCL6. BCL6 is key oncogene in MYC-driven lymphomas for which targeted treatments are currently being developed. Can

the authors comment on the interactions between BCL6 and BCOR in this setting, through DNA binding studies, and BCL6-BCOR interactions in their samples?

Response 3. We note that past observations regarding the BCL6-BCOR complex have been made in diffuse large B-cell lymphoma representing mature B-cells while the immunophenotype of Bcor mutant E μ -Myc is of a very early Pre-Pro B-cell (IgM⁻/IgD⁻, see New Supplementary Figure 14). It can therefore not be assumed that BCOR and BCL6 have similar activity in an early stage of B-cell development. BCL6 is expressed in some early Pre-B cell types, where like in the germinal centre, it is required to repress DNA damage response due to processes of light chain immunoglobulin rearrangement (Duy et al J. Exp Med. 2010 Jun 7;207(6):1209–21). It is plausible that BCOR and BCL6 may form a complex during these early stages of B-cell development but it may be equally true that other binding partners of BCOR, such as those that make up the non-canonical PRC1.1 complex of which BCOR is a member (van den Boom et al Cell Rep. 2016 Jan 12;14(2):332-46), may be important for the tumor suppressor function of BCOR. As detailed in response to **Reviewer 1, Q6** we did not find any enrichment of BCL6-BCOR complex target genes in the gene-expression data comparing BCOR mutant and WT lymphomas. We believe that further experiments to determine the putative functional importance of BCOR/Bcl-6, BCOR/PRC1.1 interactions or indeed the involvement of other BCOR binding proteins not yet identified specifically in Pre-Pro B-cells and/or E μ -Myc lymphoma is of interest but beyond the reasonable scope of this study.

Reviewer #3 (Remarks to the Author): Expert in mouse genomics

Lefebure et al.,

In their paper (Lefebure et al.,) the authors describe the genome sequencing and analysis of lymphomas from the Eu-Myc lymphoma model, which is very widely used to model human lymphoma development. The paper is well written but I have concerns about the contribution of this work as it is presented, and some of the analysis as detailed below.

Question 1. A single whole genome was generated to map the transgene. Then 23 tumor exomes were sequenced, with just two tail DNA samples used as germline controls. I would say this is far from a comprehensive analysis.

Firstly, what was the exact phenotype of these tumors? Were they thymic, splenic or did the lymphoma develop in another organ? Why sequence these tumours? What was the sample selection rationale?

Did they all have the same development/marker expression profile? (rather basic information is provided in supplementary Table 1).

Response 1. We have added further information regarding organ involvement and the immunophenotype of individual lymphomas in Supplementary Table 1. The purpose of the study was a population screen to identify novel tumor suppressors in Myc-driven lymphomagenesis therefore we made no deliberate selection of cases for sequencing.

Question 2. It is stated that two tails were sequenced for filtering and that the validation rate of the variants was +93%. Was the validation performed on the tumor DNA or using tumor AND matched normal DNA? If it's the former then there is a real potential that germline variants have slipped through the calling. There are several erroneous mouse cancer sequencing studies that have been published where match tumor/germline DNA was not used for calling. DNA "from the same colony" is not an appropriate control. For the abovementioned validation - what does a validate rate of 93% actually mean? How were these variants selected?

Response 2. To optimize the specificity and sensitivity of variant calling we used an intersection of variant callers. We have used a similar strategy in past human cancer sequencing studies (Tohill et al 2013 J. Path ; Flynn et al 2014 J. Path Wong et al Cancer Research). To reduce the chance of falsely calling private SNPs as somatic mutations we removed any variants arising in two or more closely related lymphomas. From this final filtered variant list we then randomly selected 81 variants for validation by targeted amplicon sequencing using DNA from matching lymphoma samples. This was principally done to validate the accuracy of our variant calling strategy achieving an accuracy of 92.6% as reported in the manuscript. A subset of variants (n=40) were further validated as being somatically mutated by targeted sequencing of DNA from blood taken from the matching individual animal at 4 weeks of age. We found that none of the validated somatic variants in our final filtered list could be detected in the early blood sample, which is highlighted for the candidate driver genes (n=14) in a new figure above in response to reviewer two and now is included as Supplementary Fig S5. This therefore validated our strategy of using the exome data from other lymphomas as controls.

Question 3. The indel to SNV ratio is lower than I would expect, at least based on other cancer genome studies.

Response 3. The relatively low number of InDels may be attributed to the specific mutational processes operative in E μ -Myc lymphoma or could be related to technical factors. We note that the mutation load in generally is very low in the E μ -Myc lymphoma suggesting that there are no major defects in DNA repair. We used an intersection of InDel callers to maximize specificity. Relaxing the stringency to using only a single InDel caller may have improved sensitivity (increasing the number of InDels called) but at the cost of specificity.

Question 4. The list of drivers is descriptive. No statistics are applied to this analysis. It goes without saying that *Bcor* and *Cdkn2a* are probably mutated in a significant number of cases but other genes such as *Ezh2* and *Mtor* are discussed but there is no formal proof that these genes are mutated at a frequency greater than expected by chance. Likewise for *Rpl10*.

Response 4. As indicated above, the purpose of this initial screen was to identify novel potential novel tumor suppressor and cooperating oncogenes, which we have successfully done. Rather than extend the screen even further at this stage, we chose to perform functional assays and more in-depth molecular analysis of a number of unique and interesting discoveries. First we conclusively demonstrated for the first time the tumor suppressor function of *Bcor*. Second we showed that mutation of *Cdkn2a* alone may not be sufficient to drive E μ -Myc lymphomagenesis and that tertiary co-operating mutations involving genes such as *Ras* and *Bcor* co-occur. These novel data provide new insight into the proposed tumor suppressor function of *Cdkn2a* in the context of Myc-driven malignancy. A more comprehensive analysis involving a larger number of E μ -Myc lymphomas would allow true estimation of gene mutation frequency but this is beyond the scope of this current study. As cited in the manuscript, loss of *Ezh2* has previously been shown to accelerate lymphomagenesis in the E μ -Myc model.

Question 5. My next point is on the claims surrounding *Bcor*. This gene has been described previously as being a driver in ALL, AML, T-cell prolymphocytic leukemia, sarcoma and in various other tumor types. Various mechanisms of these loss-of-function mutations have been described - Most being related to BCORs co-repressor function. The suggestion in this paper (although not proven statistically) is that Myc and *Bcor* co-operate, and this is shown functionally with shRNA and CRISPR experiments, and by expression profiling to show that *Bcor* mutant tumors have a different expression signature. This again is descriptive and there is no real attempt to define a mechanism, even using the available expression data, beyond the observation that there is TGF β pathway upregulation. The result is interesting but what does it really mean?

Response 5. We thank the reviewer for the comment and agree that detailed mechanistic insight into the tumor suppressor activity of *Bcor* would be of interest. As detailed in responses to Reviewers 1 and 2 above we have no evidence to suggest that the tumor suppressor function of *Bcor* in this model is related to its ability to interact with *Bcl6* so the question of how *Bcor* regulates the TGF β pathway and how this might affect lymphomagenesis remains open. As *Bcor* is a member of the polycomb repressor complex PRC1.1 it is possible that this is central to its tumor suppressor role and will be the subject of ongoing mechanistic studies in our laboratory. However, we believe that such extensive studies are beyond the scope of

this current manuscript and that our formal demonstration that Bcor can indeed function as a tumor suppressor protein, that the Bcor tumors have a unique gene expression profile involving the TGF β pathway and new data (new Supplementary Figure S13) showing that wild type Bcor can “rescue” the proliferative advantage conferred by mutant Bcor is a significant advance for the field. We completely agree with the reviewer that additional insight into the molecular events that underpin Bcor’s tumor suppressor activity would be of interest and we believe that the extensive analysis that we provide in this manuscript will provide sufficient information for us and others to conduct the extensive studies required to provide this information in the future.

Question 6. My final point is the relevance of the Bcor-Myc observation to human cancer. Is there a statistically significant co-occurrence of these mutations in human cancers, in particular in Myc driven lymphoma? If not how does this speak to the value of the Eu-Myc lymphoma model?

Response 6. Please see response to **Reviewer #1, Question7.**

Reviewer #1 (Remarks to the Author)

The authors have addressed my concerns in a scholarly manner.

Reviewer #2 (Remarks to the Author)

The authors have adequately addressed by concerns.

Reviewer #3 (Remarks to the Author)

I think the authors have done a reasonable job of addressing my technical concerns. I am, however, disappointed that they feel that the mechanism of how Bcor contributes to tumorigenesis in the Eu-Myc model is beyond the scope of the paper. I appreciate that such studies are hard and can take considerable time but in the absence of such insights this work does not move us significantly further forward. Maybe the TGFb pathway is important but this observation is not explored in sufficient depth for this to be proven. The expression changes described are associated with Bcor but it is not clear which (if any) of them contribute to the accelerated tumour phenotype. I feel that further work is required for this work to be published in this journal. What is presented is strong but the work is just not yet a rounded story.